# THE GRAPH LEARNING ATTENTION MECHANISM: LEARNABLE SPARSIFICATION WITHOUT HEURISTICS

## ABSTRACT

Graph Neural Networks (GNNs) are local aggregators that derive their expressive power from their sensitivity to network structure. However, this sensitivity comes at a cost: noisy edges degrade performance. In response, many GNNs include edge-weighting mechanisms that scale the contribution of each edge in the aggregation step. However, to account for neighborhoods of varying size, node-embedding mechanisms must normalize these edge-weights across each neighborhood. As such, the impact of noisy edges cannot be eliminated without removing those edges altogether. Motivated by this issue, we introduce the Graph Learning Attention Mechanism (GLAM): a drop-in, differentiable structure learning layer for GNNs that separates the distinct tasks of structure learning and node embedding. In contrast to existing graph learning approaches, GLAM does not require the addition of exogenous structural regularizers or edge-selection heuristics to learn optimal graph structures. In experiments on citation and co-purchase datasets, we demonstrate that our approach can match state of the art semi-supervised node classification accuracies while inducing an order of magnitude greater sparsity than existing graph learning methods.

## 1 INTRODUCTION

Local interactions govern the properties of nearly all complex systems, from protein folding and cellular proliferation to group dynamics and financial markets (Stocker et al., 1996; Doyle et al., 1997; Mathur, 2006; Özgür, 2011; Jiang et al., 2014). When modeling such systems, representing interactions explicitly in the form of a graph can improve model performance dramatically, both at the local and global level. Graph Neural Networks (GNNs) are designed to operate on such graph-structured data and have quickly become state of the art in a host of structured domains (Wu et al., 2019). However, GNN models rely heavily on the provided graph structures representing meaningful relations, for example, the bonds between atoms in a molecule (Fang et al., 2022). Additionally, to generate useful node embeddings, GNNs employ permutation invariant neighborhood aggregation functions which implicitly assume that neighborhoods satisfy certain homogeneity properties (Zhu et al., 2020). If noisy edges are introduced, or if the neighborhood assumptions are not met, GNN performance suffers.

To address both issues simultaneously, many GNNs include mechanisms for learning *edge weights* which scale the influence of the features on neighboring nodes in the aggregation step. The Graph Attention Network (GAT) (Veličković et al., 2017), for example, adapts the typical attention mechanism (Vaswani et al., 2017) to the graph setting, learning attention coefficients between adjacent nodes in the graph as opposed to tokens in a sequence. As we will show in Section 3, the demands of edge weighting (or structure learning) inherently conflict with those of node embedding, and edge weighting mechanisms that are joined together with node embedding mechanisms are not capable of eliminating the negative impact of noisy edges on their own.

In this paper, we introduce a method for *separating* the distinct tasks of structure learning and node embedding in GNNs. Our method takes the form of a structure learning layer that can be placed in front of existing GNN layers to learn task-informed graph structures that optimize performance on the downstream task. Our primary contributions are as follows:

1. We first introduce a principled framework for considering the inherent conflicts between structure learning and node embedding.

2. Motivated by this framework, we introduce the Graph Learning Attention Mechanism (GLAM) a layer that, when used alongside GNNs, separates the distinct tasks of structure learning and node embedding.

In addition to enabling GAT models to meet or exceed state of the art performance in semi-supervised node classification tasks, the GLAM layer induces an order of magnitude greater sparsity than other structure learning methods (Luo et al., 2020; Ye & Ji, 2021; Chen et al., 2020; Franceschi et al., 2019; Shang et al., 2021; Miao et al., 2022). Also, in contrast to the existing structure learning methods, GLAM does not employ any edge selection heuristics, exogenous structural regularizers or otherwise modify the existing loss function to accommodate the structure learning task. This makes it simpler to apply in existing GNN pipelines as there is no need to modify carefully crafted and domain-specific objective functions. Our approach is also scalable and generalizable to the inductive setting as it does not rely on optimizing a fixed adjacency matrix.

## 2 PRELIMINARIES

As our method takes inspiration from the original GAT, we begin by reviewing the mechanism by which the GAT layer generates edge weights, as well as how those edge weights are used to aggregate neighborhood information. Understanding this mechanism is important to understanding our conceptual framework (Section 3) as well as the GLAM layer (Section 4).

Graph attention networks learn weighted attention scores $e_{ij}$ for all edges between nodes $i$ and $j$, $j \in \mathcal{N}_i$ where $\mathcal{N}_i$ is the one-hop neighborhood of node $i$. These attention scores represent the importance of the features on node $j$ to node $i$ and are computed in the following manner:

$$e_{ij} = \text{LeakyReLU}\left(\vec{\mathbf{a}}^T[\mathbf{W}_{GAT}\vec{h_i} \parallel \mathbf{W}_{GAT}\vec{h_j}]\right) \tag{1}$$

where $\vec{h_i} \in \mathbb{R}^F$ are node feature vectors, $\parallel$ is vector concatenation, $\mathbf{W}_{GAT} \in \mathbb{R}^{F' \times F}$ is a shared linear transformation for transforming input features into higher level representations, and $\vec{\mathbf{a}} \in \mathbb{R}^{2F'}$ are the learnable attention weights that take the form of a single-layer feedforward neural network.

To ensure the attention scores are comparable across neighborhoods of varying size, they are normalized into attention coefficients $\alpha_{ij}$ using a softmax activation:

$$\alpha_{ij} = \text{softmax}_j(e_{ij}) = \frac{\exp(e_{ij})}{\sum_{j \in \mathcal{N}_i} \exp(e_{ij})} \tag{2}$$

For stability and expressivity, the mechanism is extended to employ multi-head attention, and the outputs of the $K$ heads in the final layer are aggregated by averaging:

$$\vec{h}'_i = \text{softmax}\left(\frac{1}{K}\sum_{k=1}^{K}\sum_{j \in \mathcal{N}_i}\alpha_{ij}^k\mathbf{W}_{GAT}^k\vec{h}_j\right) \tag{3}$$

In the next section, we explain why the normalization procedure in Eq. 2, while crucial for node embedding, is an impediment for structure learning.

## 3 CONFLICTING DEMANDS: NODE EMBEDDING VS. STRUCTURE LEARNING

At first glance, it may seem we could address the structure learning problem by simply thresholding the existing GAT attention coefficients $\alpha_{ij}$. However, due to the need for neighborhood-wise normalization and permutation invariant aggregation, this would not be ideal. As a motivating example, consider the following: if we add three random edges per node to the standard Cora dataset McCallum et al. (2000), then train a GAT to perform semi-supervised node classification, we get

a classification accuracy of 65.9%. If we manually set the edge weights for each of those newly added random edges to zero (note: the GAT is not able to learn this weighting on its own) the same GAT is able to achieve 79% accuracy. However, if we simply remove all the newly added random edges so the GAT does not consider them at all, the GAT can achieve 82.1% accuracy. In GNNs with permutation-invariant aggregation mechanisms, dropping noisy edges is more effective than learning to zero their edge weights.

To understand this result, it's important to understand why the GAT attention coefficients are calculated as the softmax of the attention scores. This softmax step serves two important purposes:

1. It normalizes the attention scores into attention coefficients that sum to one, which ensures the sum of the neighboring representations (as defined in Eq. 3) is also normalized. This is a crucial function of any node embedding mechanism because it normalizes the distributional characteristics of $\vec{h_i'}$ not just across different neighborhoods but across neighborhoods of varying size. Without this, the downstream layers that ingest $\vec{h_i'}$ would need to account for widely variable magnitudes in the values of $\vec{h_i'}$, and performance would suffer.

2. The softmax serves as an implicit, low-resolution structure learning device. To locate the maximum input element in a differentiable manner, softmax uses exponentials to exaggerate the difference between the maximum element and all of the rest. In graph attention networks, this means exaggerating the difference between the attention coefficient of the *single most important neighbor* vs. all the rest. This imbues the attention coefficients, and thus each nodes' neighborhood, with a soft-sparsity that improves the learned node embeddings by minimizing the influence of all but the single most useful neighbor.

For these reasons, if our aim is to generate useful node embeddings for node-wise prediction, we should not do away with the softmax activation to normalize attention coefficients in each neighborhood. However, as our aim is to jointly learn useful node embeddings *and* the graph structure, this embedding mechanism alone is not sufficient. A distinct structure learning mechanism would require the following:

- To learn the graph structure, we need to assess the value of each neighbor *independently*. This is because, in graph learning, local neighborhoods are subject to noisy evolution as the network samples edges. If the value of each neighbor is conditional on the present neighborhood, it will be difficult to disentangle the relative value of one neighbor from another as the neighborhood evolves.

This is why the existing edge-weighting + node embedding paradigm is insufficient if our goal is to simultaneously learn node embedding and graph structure: the neighborhood-wise normalization (softmax over edge-weights) expresses each neighbors' importance *relative* to all the other nodes in the neighborhood, which is in direct conflict with the edge-wise independence requirements of structure learning.

To preserve the embedding advantages of GNNs while accommodating the conflicting demands of structure learning, we introduce the Graph learning Attention Mechanism (GLAM).

## 4 THE GRAPH LEARNING ATTENTION MECHANISM (GLAM)

Like the GAT layer, the GLAM layer ingests the node features $\mathbf{h} \in \mathbb{R}^{N \times F}$ and the edge set $\mathcal{E}$, where $N$ is the number of nodes and $F$ is the number of features per node. For each node $i$, we transform the node features $h_i$ into higher order representations $x_i$ using a shared linear transformation $\mathbf{W} \in \mathbb{R}^{F_S \times F}$:

$$x_i = \mathbf{W}(\vec{h_i}), \, x_i \in \mathbb{R}^{F_S} \tag{4}$$

For each edge between nodes $i$ and $j$ in the provided edge set $\mathcal{E}$, we then construct a representation for that edge by concatenating the node representations $x_i$ and $x_j$. From here, our GLAM layer differs from the GAT layer. Using another shared linear layer $\mathbf{S} \in \mathbb{R}^{1 \times F_S}$, we map these

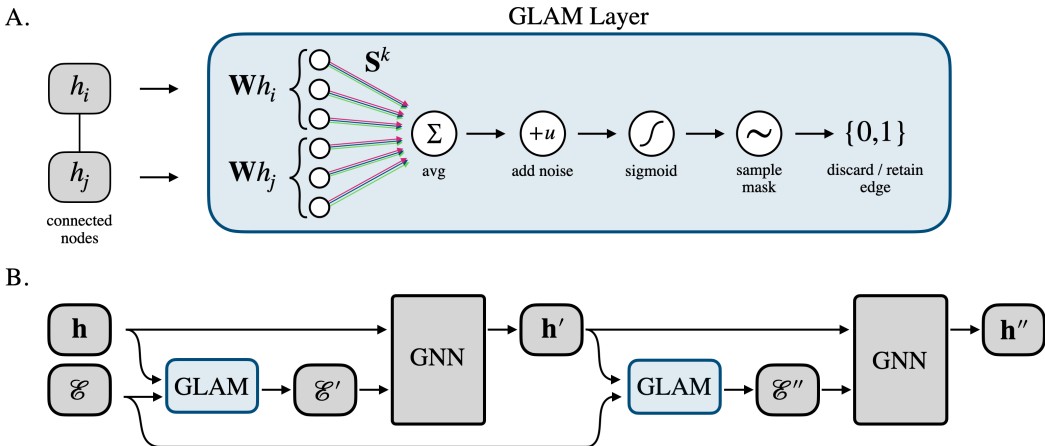

Figure 1: **A**: An illustration of the Graph Learning Attention Mechanism operating on a single pair of connected nodes with features $h_i, h_j \in \mathbb{R}^F$, using the shared weight matrix $\mathbf{W} \in \mathbb{R}^{F_S \times F}$ and multi-headed attention with $K = 3$ heads. **B**: A high level overview of how the GLAM layer is used to learn optimal graph structures at each layer. The input is the original graph with node features $\mathbf{h} \in \mathbb{R}^{N \times F}$ and edge set $\mathcal{E}$. To make as few assumptions about the optimal graph structure as possible, we feed the original edge set $\mathcal{E}$ into each GLAM layer to reassess the utility of each edge at each layer. We note that this is optional, and chaining edge sets across layers is also possible.

edge representations onto structure learning scores $\boldsymbol{\eta} \in \mathbb{R}^{|\mathcal{E}| \times 1}$, where the score $\eta_{ij}$ for each edge becomes:

$$\eta_{ij} = \sigma\Big(\mathbf{S}\big[x_i \| x_j\big] + u\Big) \tag{5}$$

where $\|$ is vector concatenation, $u \in \mathbb{R}^1$ is i.i.d. noise drawn from a $\mathrm{U}(-0.5, 0.5)$ distribution centered about zero, and $\sigma$ is a sigmoid activation allowing each $\eta_{ij}$ to be interpreted as the probability of retaining the edge between nodes $i$ and $j$. We note here that the noise term is not strictly required, but we found that it helped to smooth out the training.

Finally, we extend the GLAM layer to include $K$ attention heads, and the final structure learning score $\eta_{ij}$ for each edge becomes:

$$\eta_{ij} = \sigma\Big(\frac{1}{K} \sum_{k=1}^{K} \mathbf{S}^k\big[x_i \| x_j\big] + u\Big) \tag{6}$$

Next, we sample a discrete mask $M \in \{0, 1\}^{|\mathcal{E}|}$ from the distribution parameterized by the structure learning scores $\boldsymbol{\eta}$. When applied to the given graph $\mathcal{E}$, we get a sparsified graph $M(\mathcal{E}) \rightarrow \mathcal{E}' \subseteq \mathcal{E}$. To sample the discrete values in $M$ from the continuous probabilities in $\boldsymbol{\eta}$ differentiably, we use the Gumbel-Softmax reparameterization trick introduced by Jang et al. (2016) and retain the edge between nodes $i$ and $j$ if $\eta_{ij} > 0.5$.

This new graph $\mathcal{E}'$ is then used in place of $\mathcal{E}$ in the downstream GNN representation learner. If that downstream GNN were a GAT, using the equations from Section 2, the attention coefficients would become:

$$\alpha_{ij} = \text{softmax}_j(e_{ij}) = \frac{\exp(e_{ij})}{\sum_{j \in \mathcal{N}_i'} \exp(e_{ij})}$$

and (7)

$$\vec{h}_i' = \text{softmax}\left( \frac{1}{K} \sum_{k=1}^{K} \sum_{j \in \mathcal{N}_i'} \alpha_{ij}^k \mathbf{W}_{GAT}^k \vec{h}_j \right)$$

where the neighborhoods $\mathcal{N}_i'$ are defined by the non-masked edges in $\mathcal{E}'$ where $\boldsymbol{\eta} > 0.5$.

As each GNN model incorporates graph structure in its own way, we have ensured that GLAM produces a maximally general, differentiable mask on the given edges. This mask may be readily used to separate the structure learning tasks from node embedding regardless of the particular embedding mechanism.

For example, in our implementation, we incorporate the learned graph $M$ into the GAT by trivially swapping the softmax over the attention coefficients with a sparse softmax that respects the masked edges (as shown in Eq. 7). Extending beyond the GAT, we could incorporate the learned graph into the widely used Graph Convolutional Network (GCN) (Kipf & Welling, 2016), for example, by simply multiplying the adjacency matrix $A$ by the mask $M$ before the application of the renormalization trick. To apply $M$ in arbitrary GNNs, we need only to insert it before the neighborhood aggregation step.

## 5 RELATED WORK

The structure learning literature includes many effective methods for inducing task-informed, sparsified subgraphs. However, each of them relies on either edge-selection heuristics (such as top-k selection) or exogenous structural regularizers (such as penalties on retained edges) to stabilize the structure learning process. In contrast, our approach requires none of these, making it more readily deployable in existing GNN pipelines and obviating the need to interfere with carefully crafted objective functions or training methodologies.

Approaches such as Neural Relational Inference (Kipf et al., 2018) and Graphs for Time Series (Shang et al., 2021) introduce methods for learning probabilistic graph models optimized for various time-series forecasting tasks. However, while both induce sparse subgraphs, they rely on exogenous regularization (penalties on retained edges) and suffer from scalability issues as both learn fixed adjacency matrices.

More closely related to our approach are the Graph Stochastic Attention, GSAT (Miao et al., 2022) and Stochastic Graph Attention, SGAT (Ye & Ji, 2021) models, which also adapt the graph attention layer to learn sparse subgraphs. SGAT attaches a binary gate to each edge in the given graph, then learns a single adjacency matrix that is shared across all layers. While effective, this adjacency matrix makes it difficult to scale and they employ an $L_0$ norm in the loss function to penalize retained edges.

GSAT (Miao et al., 2022), based on the graph information bottleneck principle (Wu et al., 2020) is designed to be an interpretable graph learner and uses a similar sparse attention mechanism to assess the utility of each edge. However, in spite of not using sparsity constraints such as $L_0$ norms, they do add additional structural regularization terms into the loss function. Additionally, learnable weights in GSAT are shared between the GNN used to generate the binary mask and the GNN used for downstream node embedding. As we reviewed in Section 3, the demands of structure learning conflict with those of node embedding, so sharing weights/architectures between these networks may reduce performance in some tasks.

As GSAT was not evaluated on our datasets, and SGAT more closely resembles GLAM in design and functionality (inducing an edge sparsified graph) we confine our direct comparison to SGAT to ensure fairness and clarity.

|  | Type | Nodes | Edges | Features | Classes | $H(G)$ |
|---|---|---|---|---|---|---|
| **Cora** | Citation Network | 2,708 | 13,264 | 1,433 | 7 | 0.83 |
| **PubMed** | Citation Network | 19,717 | 108,365 | 500 | 3 | 0.79 |
| **Citeseer** | Citation Network | 3,307 | 12,431 | 3,703 | 6 | 0.71 |
| **Amazon Photo** | Co-Purchase | 7,487 | 245,812 | 745 | 8 | 0.84 |
| **Amazon Computers** | Co-Purchase | 13,385 | 505,474 | 767 | 10 | 0.79 |

Table 1: Homophilic graph datasets used in our experiments.

## 6 EXPERIMENTS

To demonstrate the efficacy of the GLAM layer, we report performance in semi-supervised node classification tasks on real-world graph datasets. As it is well known that GNNs performance is degraded by heterophilic graphs (in which adjacent nodes tend to have different labels) we look only at those graphs where the edges bring material value for GNNs, i.e. homophilous graphs. Formally, the edge homophily ratio of a graph is a value the range [0, 1] that denotes the fraction of edges in the graph that join nodes with the same labels:

$$H(G, \{y_i; i \in \mathcal{V}\}) = \frac{1}{|\mathcal{E}|} \sum_{(i,j) \in \mathcal{E}} \mathbb{1}(y_i = y_j) \tag{8}$$

Where $G$ is an input graph with nodes $v \in V$ having labels $y$. While homophily isn't an unrealistic assumption, with most graphs being constructed this way (McPherson et al., 2001), a method for graph-adaptation based on the assumptions of the downstream GNN is still desirable as many GNN models implicitly assume a high degree of homophily and perform poorly when this assumption is violated (Zhu et al., 2020). In graphs with low homophily, such as the WebKB[1] datasets, GNN performance is often optimized by removing nearly every edge. For this reason, to get a better understanding of the GLAM layer's efficacy, we confine our study to the homophilic graphs detailed in Table 1.

Cora, PubMed and Citeseer (McCallum et al. (2000), Sen et al. (2008), Giles et al. (1998)) are citation datasets with relatively low average degree. Nodes correspond to documents (academic papers) and edges represent citations between these documents. Node features are the bag of words representation of the document and the task is to classify each document by its topic. Amazon Photo and Amazon Computers (Shchur et al. (2018)) are co-purchase datasets with a much higher average degree, where each node corresponds to a product and two nodes are linked if those products are frequently bought together. The node features are also a bag of words representations but of the reviews of each product. Similarly, the task is to classify each product into its product category.

For a fair comparison with SGAT, we use the same canonical splits in each dataset: 20 nodes per class for training, 500 nodes for validation and 1,000 for testing.

### 6.1 TRAINING METHODOLOGY

In all our experiments, we train a two layer GAT, using the GLAM layer to learn optimal structures for each GAT layer. Crucially, to demonstrate how the GLAM layer may be 'dropped in' to an existing GNN model with little to no modification to the training scheme, we use the known optimal hyperparameters and training methodology for the GAT layers (as reported in Veličković et al. (2017)) without modification. This covers all components of the model including loss functions, regularization, optimizers and layer sizes. The only hyperparameters we modify are those associated with the inserted GLAM layers and, as mentioned in Section 4, the learned mask is enforced using a sparse softmax activation in the GAT layers that respects the mask $M$. As the Co-Purchase datasets were not tested in the original GAT paper, we began optimization with the same hyperparameters as

---

[1]http://www.cs.cmu.edu/afs/cs.cmu.edu/project/theo-11/www/wwkb/

| Dataset | Cora | PubMed | Citeseer | Photo | Computers |
|---------|------|--------|----------|-------|-----------|
| MLP | 49.5% | 68.1% | 46.9% | 70.5% | 53.1% |
| GAT | 82.4% | 78.2% | 70.9% | 89.1% | 81.5% |
| SGAT | 83.0% | 78.3% | 71.5% | 89.9% | 81.8% |
| GLAM | 82.4% | 78.6% | 70.5% | 90.3% | 83.0% |
| % Edges Removed (layer 1 / layer 2) | | | | | |
| SGAT | 2.0 / − | 2.2 / − | 1.2 / − | 42.3 / − | 63.6 / − |
| GLAM | 22.3 / 2.2 | 30.0 / 0.3 | 46.3 / 1.4 | 63.1 / 13.2 | 55.0 / 8.7 |

Table 2: Top: Semi-supervised node classification accuracies are listed in percent. Bottom: Percentage of edges removed at the first / second layer. For SGAT, a single graph is learned at the first layer and reused it for all following layers. In our experiments, the GLAM layer is used to learn an optimal graph at each layer.

Cora, and found that increasing the hidden dimension from 8 to 32 led to optimal performance. In all cases, we use cross entropy as our loss function and the Adam optimizer Kingma & Ba (2014) to perform gradient descent. In all experiments we report the average classification performance and induced sparsity over 10 independent trials (Table 2). All hyperparameters are available in the GitHub.

Additionally, to make as few assumptions about the optimal graph as possible, we use the GLAM layer to independently assess the utility of each edge at each layer, as shown in Figure 1B, and report the induced sparsity at each layer. This is in contrast to existing methods which learn a single graph at the first layer that is then reused for all the downstream layers. As we will discuss more in Section 6.2, learning the optimal graph at each layer not only improves performance and induces greater sparsity but allows us to better observe the relationship between the data, the GNN aggregator, and the downstream task more clearly.

We note here that the use of exogenous regularizers and edge-selection heuristics, as employed by other structure learning methods, is to help the structure learner better adapt to the demands of the downstream GNN. With the GLAM layer, we enable this same sort of adaptation by simply making these layers more sensitive than the downstream GNN layers. To do so, we relax the regularization (weight decay) and amplify the learning signal (increase the learning rate) for just the GLAM layers. Doing so for only the GLAM layers allows them to adapt to the GNN representation learner without the need for additional terms in the loss function. These layer-wise changes are slight and optimal values vary by dataset. All layer-specific learning rates and regularization coefficients are outlined in the hyperparameter configuration files at [GitHub redacted for review].

Finally, as self-loops have disproportionate utility in any node-wise prediction task, the GLAM layers only attends over those edges that adjoin separate nodes, retaining all self-loops by default.

## 6.2 PREDICTION PERFORMANCE AND INDUCED SPARSITY

On semi-supervised node classification datasets, the GLAM layer enables the GAT to reach similar accuracies while inducing an order of magnitude greater sparsity than existing methods. This increased sparsification is notable as we do not explicitly enforce sparsity constraints or penalize retained edges. As the GLAM method is fundamentally about inducing task-informed subgraphs, and classification accuracies are all similar to SGAT, we confine our analysis to the sparsification aspect of the results. Complete results can be found in Table 2.

On each of the citation datasets, the GLAM layer is competitive with SGAT on prediction performance while inducing over an order of magnitude greater sparsity. This degree of sparsity has not yet been induced in these datasets while preserving SOTA classification performance. As such, we view these results as indicating, in addition to the efficacy of the GLAM layer, that there may be more redundant edges in these datasets than previously thought. Part of the advantage of the GLAM

layer is that we can train it using using only the signal from the downstream task. As such, the induced graph is a reflection purely of the relationship between the data, the GNN aggregation scheme, and the downstream task. Following from this, we note that the first GLAM layer trained on these datasets removes closer to $1 - H(G)$ percent of the edges. There is not an exact correspondence, but the proximity between these two quantities is likely a reflection of the GLAM layers learning that the downstream GAT performs best on neighborhoods with higher homophily. The second GLAM layer removes on the order of 1% of the edges, which is similar to SGAT. We hypothesize that more edges are retained in the second and final layer due to the node representations already containing information from their 1-hop neighborhood, and there being additional value in aggregating information from the 2-hop neighborhood.

On the co-purchase datasets which have much larger average degrees, the GLAM layers remove substantially more edges in the first layer but fewer in the second. We emphasize here that as there are no structural regularizers, GLAM is not encouraged to learn the sparsest graph possible, but rather the graph which optimizes downstream task performance. As we can see with the Computers dataset, retaining a few more edges in the first layer and a few less in the second resulted in substantially higher classification accuracies.

While our method does not always achieve state of the art classification performance, we do achieve similar performance with far greater sparsification and without changes to the canonical loss function. As the GLAM layer is differentiable and can be readily integrated with arbitrary downstream GNNs, we believe it could be widely useful and is worthy of further study.

## 7 CONCLUSIONS

We presented a principled framework for considering the graph structure learning problem in the context of graph neural networks, as well as the Graph Learning Attention Mechanism (GLAM) a novel structure learning layer motivated by this framework. In contrast to existing structure learning approaches, GLAM does not require exogenous structural regularizers nor does it utilize edge-selection heuristics. In experiments on citation and co-purchase datasets the GLAM layer allows an unmodified GAT to match state of the art performance while inducing an order of magnitude greater sparsity than other graph learning approaches.

There are several improvements that could be made to the method, especially as it relates to stabilizing the structure learning layers without augmenting canonical loss functions. Future research could be directed at adaptive methods of decaying the learning rates in those layers or layer-specific early stopping.

Finally, as our approach yields graph structures uncorrupted by the influence of exogenous heuristics, we also see the potential for its use in the design and analysis of novel GNN architectures. In the GLAM layer, edges are retained or discarded based whether the downstream GNN can make productive use of them. By interpreting the distributions of retained edges at each layer, designers could, for example, better understand how well some aggregation scheme integrates information from heterophilic vs. homophilic neighborhoods. In a similar way, GLAM may be used as a principled method for assessing the value of depth in GNNs, a persistent issue due to the over-smoothing problem (Oono & Suzuki, 2019). If a GNN layer were to no longer benefit from the incorporation of structural information, i.e. the number of retained edges yielded by its preceding GLAM layer was close to zero, then the GNN may be too deep, and adding additional layers may introduce complexity without increasing performance.

## 8 REPRODUCIBILITY

All code, data and configuration scripts are available in [GitHub redacted for review]. Our models were built using the open source PyTorch Geometric library (Fey & Lenssen, 2019) and we provide setup scripts to configure an Anaconda environment that contains all necessary packages. All hyperparameters and the training methodology are clearly documented in the code and the benchmarks are runnable on CPU or GPU with a single command.

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
