# OpenReview forum: "The Graph Learning Attention Mechanism: Learnable Sparsification Without Heuristics"
_ICLR.cc/2023/Conference — Submitted to ICLR 2023_

### Official Review · Reviewer_jLy4 · 2022-10-21

**Confidence:** 4
**Correctness:** 2
**Technical Novelty And Significance:** 2
**Empirical Novelty And Significance:** 2
**Recommendation:** 3

**Clarity, Quality, Novelty And Reproducibility:**

Overall, this paper presents an attention mechanism slightly different from the Softmax attention as the graph structure learner extracts sparse subgraphs to be effective topological information to facilitate node embedding learning. I like the idea of disentangling the learning of graph structure and embeddings, but the analysis is less insightful and lacks more principled discussion in Sec. 3. Also, I do not see significant breakthroughs and novelty for the composition of the GLAM layer. For the claimed advantages, there are no substantial improvements or clear justifications in experiments.

**Strength And Weaknesses:**

Pros:
- The insight of the addressing the conflicts between structure learning and node embedding optimization is interesting
- Simplicity of the methodology
- The paper is well written and easy to follow despite some confusing points

Cons:
- Lack of a more principled motivations and justifications behind the proposed design
- Limited evaluation setup
- Limited technical novelty


**Summary Of The Paper:**

This work presents a graph structure refinement layer named GLAM based on a neighborhood attention schema slightly different from the Softmax attention, which disentangles the optimization of node embeddings and graph structures. The proposed GLAM layer learns more sparse subgraphs than prior arts by giving the known adjacency matrix.

**Summary Of The Review:**

Following my comment above, there are several questions:

- What are the exact conflicts between the learning of graph structures and node embeddings? Specifically, in Sec. 3, I have these questions: (1) What are the roles of the permutation invariant aggregator in hindering the use of $\alpha_{ij}$ as the bases of structure learning? (2) I partially agree with your analysis about the Softmax attention, but how to get your conclusion of “For these reasons, …, we should not do away with the Softmax activation to normalize attention coefficients…”? (3) I cannot see your principled analysis to answer why it is not desired to learn edge weights together with node embeddings, e.g., how do you estimate the noisy evolution of local neighbourhoods, and why the improvements over GAT are less significant if your conjectures hold?

- I notice the authors mentioned graph sparsity several times, especially in Sec. 6, but there is no experiment to discuss the consequent benefits, e.g., efficiency and scalability. Have you evaluated the proposal on some larger real-world graphs, e.g., OGB datasets?

---

> ### Author Response · Authors · 2022-11-05
> **Reply to Reviewer jLy4**
>
> Dear Reviewer,
>
> Thank you very much for your thoughtful review. Below, we address the concerns you raised in your review.
>
> 1. For a more principled motivation, please see our addition to Section 3: If we add three random edges per node to the standard Cora dataset, then train a GAT to perform semi-supervised node classification, we get a classification accuracy of 65.9%. If we manually set the edge weights for each of those newly added random edges to zero (note: the GAT is not able to learn this weighting on its own) the same GAT is able to achieve 79% accuracy. However, if we simply remove all the newly added random edges so the GAT does not consider them at all, the GAT can achieve 82.1% accuracy. In GNNs with permutation-invariant aggregation mechanisms, dropping noisy edges is more effective than learning to zero their edge weights. We have updated Section 3 to include this motivating example.
>
> 2. Permutation invariant aggregators must use a permutation invariant aggregation function (typically addition) to aggregate the neighboring node representations. In order to normalize the distributional qualities of these representations across nodes with varying numbers of neighbors, we need to scale each neighbor’s representation so that the aggregation of all the neighbors yields a representation that is comparable across neighborhoods of different size. This is the role of the softmax, and as such we cannot do without a softmax or similar activation function that will normalize the edge weights.
>
> 3. Following from this, the reason we cannot use softmax for structure learning is because the softmax (defined in Eq 2) exponentiates the attention scores e_ij before normalizing the sum of the exponentiated neighbors. Thus, the only way to generate an attention _coefficient_ for edge ij would be for the attention score e_ij to be -infinity. In short: even an attention score of zero will yield a positive normalized attention coefficient.
>
> 4. We believe that induced sparsity can be interesting and valuable outside of computational/scalability concerns. In our experiments, for example, we believe that our method demonstrates that there are more noisy edges than previously believed in widely used citation datasets. This on its own is an interesting result.
>
> As a final note, please see our clarification in Sections 2 and 4: we are not simply replacing the softmax with a slightly different activation, but instead introducing a separate structure-learning layer with its own learnable weights. These are distinct from the attention weights and linear layers present in the GAT. This has been clarified in Sections 2 and 4 where we have explicitly denoted the learnable weights in the GAT as W_{GAT} and the learnable weights in the GLAM layer as W.
>
> Thank you again for your careful review, and please let us know if you have any remaining questions.

---

### Official Review · Reviewer_aSBH · 2022-10-23

**Confidence:** 3
**Correctness:** 3
**Technical Novelty And Significance:** 3
**Empirical Novelty And Significance:** 3
**Recommendation:** 5

**Clarity, Quality, Novelty And Reproducibility:**

This paper is well-written and organized with clearly defined objectives and motivations.

This work builds on existing methods including Graph Attention Networks (GAT). In breaking down the GLAM method, the similarity and novelty of this method from GAT are clearly noted.

The authors take note of the constraints on the datasets that would benefit from applying the GLAM layer for training.

Source code and setup scripts are provided but redacted for review.


**Strength And Weaknesses:**

The experiments indicate the GLAM layer when applied achieves comparable prediction accuracies to state-of-the-art methods on benchmark datasets but with a significantly sparser graph structure.

Since the experiments do not show great improvement in prediction accuracies on benchmark datasets, the justification for using this method would rely on applications that can take full advantage of the edge removal capabilities in this paper.

Future experiments can uncover the reasoning behind the drastic differences in the number of removed edges between the first and second GLAM layers.

During experiments, each dataset has a drastically different % of edges removed during training. Further research may be able to explain this occurrence.

Determining cutoffs for low homophily thresholds for usable datasets can open the door for applications to more diverse datasets and applications.

Does the GLAM layer offer any runtime benefits during training?


**Summary Of The Paper:**

This paper addresses issues of GNN sensitivity to graph structures. Specifically how noisy edges degrade performance which cannot be mitigated without complete removal. This paper proposed a novel layer Graph Learning Attention Mechanism (GLAM) which isolates structure learning from node embeddings. This technique does not require using costly exogenous structural regularizers or edge-selection heuristics to learn optimal graph structures. Experiments on benchmark datasets including Cora, PubMed, Citeseer, Amazon Photo, and Amazon Computers show comparable classification performance with significantly higher sparsity in structure.

**Summary Of The Review:**

The work in this paper is good with promising results in an interesting direction beyond plainly increasing prediction accuracy on benchmark datasets. Additional work may be needed to expand upon this research for publication.

---

> ### Author Response · Authors · 2022-11-05
> **Reply to Reviewer aSBH**
>
> Dear Reviewer,
>
> Thank you very much for your thoughtful review. Below, we address the concerns you raised in your review.
>
> 1. We disagree that the justification for inducing higher degrees of sparsity relies on computational/scalability concerns alone. In our experiments, for example, we believe that our method demonstrates that there are more noisy edges than previously believed in widely used citation datasets. This on its own is an interesting result.
>
> 2. We agree and are working on better understanding why so many more edges are removed in the first layer. We will update this revision and notify reviewers once that is ready.
>
> 3. While each dataset yields different sparsity results, this is expected. Each of these datasets was collected and curated independently, and as such likely contain varying amounts of noisy edges. We do not view these differences in induced sparsity as a weakness of the proposed method.
>
> 4. During training, there are not any substantial benefits to the runtime from the introduction of GLAM, as the GLAM layer attends over all the given edges. However, it does not do so by optimizing some adjacency matrix, and as such does not necessarily scale like N^2, but instead by the number of given edges E.
>
> Thank you again for your careful review, and please let us know if you have any remaining questions.

---

### Official Review · Reviewer_7cxi · 2022-10-25

**Confidence:** 3
**Correctness:** 2
**Technical Novelty And Significance:** 2
**Empirical Novelty And Significance:** 3
**Recommendation:** 3

**Clarity, Quality, Novelty And Reproducibility:**

Clarity: Good

Quality: Fair
- I think the evaluation of the proposed method is not good enough, which makes the quality of this work fair. Please refer to the weakness for more details.

Novelty: Fine (between Fair and Good)
- I appreciate the observation that the structure learning and node embedding, as claimed in the paper, this has never been done before. I feel this looks like the major novelty of this work.
- The proposed method looks very similar to GAT, but replacing the LeakyReLU in eq.(1) and softmax in eq.(2) with the plus noise and sigmoid in eq.(5), I don't think this part has enough novelty.

Reproducibility: Fair
- Please refer to the weakness for more details.


**Strength And Weaknesses:**

Strength:
1. This work is clearly motivated, and the studied topic is meaningful.
2. The writing of this work is easy-to-follow.

Weakness:
1. Model Evaluation
1) The proposed GLAM is only examined with 3 baselines, which I think is not enough.
- The authors mentioned that "GSAT was not evaluated on our datasets", I am wondering why GSAT is excluded.
- I think it would be better if the authors can also compare with more challenging baselines, especially some SOTA models on structure learning.
2) no ablation study is provided to examine the GLAM method comprehensively, for some examples, I am curious about the following aspects:
- is it necessary to apply a GLAM layer before every GNN layer?
- the authors claim that the GLAM can be incorporated with various GNN backbones, however, only an example with GAT is provided. I am curious to see how GLAM performs on other GNN models, such as GCN.

2. Reproducibility
- Though the authors provide the section to clarify the reproducibility and the experimental set-up details claimed to be provided in the code, the code link is missing. So essentially no useful information is provided in the paper, which makes the reproducibility fair.


Some questions:
1. In equations (5) and (6), what is the major benefit of adding the noise term u? is it for the Gumble softmax trick for differentiability?
2. if we apply GAT after GLAM, it seems to me that we are still jointly considering the structure learning and node embedding tasks simultaneously as the model still uses the weight on each edge learned by GAT for message passing. Does it contradict the claim that these two objectives should be considered separately?
3. It looks to me that, the major drawback of GAT is the softmax which expresses the neighbors' importance relatively to other nodes. Then, does it mean, as long as we come up with a way to get rid of this softmax, the limitation can be solved?
4. In section 5, I don't quite understand, why it would be problematic to add penalties on retained edges?
5. In section 6.2, the authors claims one of the benefits of GLAM is sparsification, I am wondering, is this sparsification mainly for efficiency purpose? If so, then how much speed/memory improvement can this sparsification bring to GAT? If not, then why is sparsification important here?

**Summary Of The Paper:**

This work aims to tackle the challenges of noisy edges in graph learning problems. It first points out the finding that the demands of structure learning and node embedding are inherently conflicting with each other, so it is not proper to optimize both jointly.  Then, based on their finding, they propose the Graph Learning Attention Mechanism （GLAM） layer, which is a non-heuristic differentiable network layer that can be inserted into existing GNN models and can focus on the graph structure learning objective to learn a task-informed graph structure and benefit the graph learning.

**Summary Of The Review:**

This is a well-motivated work focusing on how to separate the structure learning and node embedding task to best solve the graph learning problems. Though the writing is clear and the explored topic is meaningful, I have major concerns about the method evaluation. First, it is not compared with comprehensive suitable baselines, and the benefit of the proposed method is not distinctive; second, the authors only provide one simple experimental result (in table 2), and no ablation studies are provided to examine the GLAM method comprehensively; another minor point is, neither the code link nor details of the experimental set-up are provided.

---

> ### Author Response · Authors · 2022-11-05
> **Reply to Reviewer 7cxi**
>
> Dear Reviewer,
>
> Thank you very much for your thoughtful review. Below, we address the concerns you raised in your review.
>
> 1. We do not compare with GSAT to ensure a fair comparison. GSAT was not originally evaluated/optimized on the citation and co-purchase datasets we used to validate our method, while SGAT was, and SGAT bears slightly more resemblance to our method. This is mentioned in the last paragraph of Section 5. Additionally, we confine our comparison to GAT and GAT-based techniques as that is the downstream GNN we have chosen in our experiments. This again is to demonstrate the efficacy of the GLAM layer at inducing useful sparsity towards optimizing a downstream objective.
>
> 2. It is not necessary to learn a new graph at each layer, but we have done so to ensure maximum generality and to demonstrate the ease of placing the GLAM layer at arbitrary positions within an existing network. We are working on additional ablations now and will update the paper and this rebuttal when ready, including GCN and GLAM+GCN.
>
> To address your questions:
>
> 1. We note that the noise term is not strictly required, but we found that it helped to smooth out the training. We have added this clarification to Section 4.
>
> 2. Using GAT after GLAM does not contradict the proposed claim that these two models have complementary strengths. The GLAM layer is filtering out the noisy edges while the GAT layer is embedding the nodes given the remaining edges, which brings us to your third question:
>
> 3. As mentioned in Section 3, we do not want to get rid of the softmax on the edge weights in the GAT as that softmax is what enables us to compare neighborhoods of different sizes. This softmax is critical to normalize the distributional characteristics of the node embeddings and cannot be done away with without substantial changes to the GAT layer.
>
> 4. It would not necessarily be problematic if all we were concerned with was task performance, but exogenous regularizers such as penalties on retained edges necessarily corrupt the training signal in a non-task-informed manner. Using penalties on retained edges, the final topology with then be a function of the input data + GNN + task + exogenous regularizers instead of simply input data + GNN + task.
>
> 5. In our case, we do not explicitly enforce sparsity constraints and yet the model learns to drop an order of magnitude more edges than other techniques. This is an advantage in these settings because it is essentially able to filter out more noisy edges than any other methodology. The magnitude of induced sparsity was a surprising result on these particular datasets as there have been many approaches that have not achieved even close to this degree of filtering.
>
> As a final note, we are not replacing the LeakyReLU in eq.(1) and softmax in eq.(2) with the plus noise and sigmoid in eq.(5) but instead introducing a separate structure-learning layer with its own learnable weights. These are distinct from the attention weights and linear layers present in the GAT. This has been clarified in Sections 2 and 4 where we have explicitly denoted the learnable weights in the GAT as W_{GAT} and the learnable weights in the GLAM layer as W.
>
> Thank you again for your careful review, and please let us know if you have any remaining questions.

---

### Official Review · Reviewer_BqHK · 2022-10-30

**Confidence:** 4
**Correctness:** 3
**Technical Novelty And Significance:** 2
**Empirical Novelty And Significance:** 2
**Recommendation:** 5

**Clarity, Quality, Novelty And Reproducibility:**

This paper is well-written. Topology optimization is not new in graph learning, especially edge movement. The paper points out a conflict between node embedding and structure learning, which seems new but not convincing.

**Strength And Weaknesses:**

Strengths,

S1: The proposed topology optimization method is differentiable;

S2: This paper studies an interesting problem;

S3: This paper is well-written.

Weaknesses,

W1: There is a lack of evidence to support the analysis of the conflations in Section 3, which weakens the motivation of this paper.

W2: Topology optimization is not new in graph learning, especially edge movement.

W3: The improvement from GLAM seems limited in experiments. And the empirical support for the effectiveness of GLAM is weak.


**Summary Of The Paper:**

This paper proposes a topology optimization method for message passing-based graph learning models, termed Graph Learning Attention Mechanism (GLAM). The proposed model is differentiable and removes edges in an end-to-end manner. The method can be integrated into GCN and GAT. The experimental results show the proposed GLAM can improve the performance of GCNs.

**Summary Of The Review:**

I recommend a weak reject because of the weak motivation and validation.

---

> ### Author Response · Authors · 2022-11-05
> **Reply to Reviewer BqHK**
>
> Dear Reviewer,
>
> Thank you very much for your thoughtful review. Below, we address the concerns you raised in your review.
>
> 1. As a motivating example, consider the following: if we add three random edges per node to the standard Cora dataset, then train a GAT to perform semi-supervised node classification, we get a classification accuracy of 65.9%. If we manually set the edge weights for each of those newly added random edges to zero (note: the GAT is not able to learn this weighting on its own) the same GAT is able to achieve 79% accuracy. However, if we simply remove all the newly added random edges so the GAT does not consider them at all, the GAT can achieve 82.1% accuracy. In GNNs with permutation-invariant aggregation mechanisms, dropping noisy edges is more effective than learning to zero their edge weights. We have updated Section 3 to include this motivating example.
>
> 2. While topology optimization is not new, all such methods include some exogenous regularizers, usually either as penalties on retained edges or as edge selection heuristics such as top-k selection. Our method is an attempt to solve the topology optimization problem without exogenous regularizers, using only the information from downstream task performance to optimize the graph. We mention this in the final paragraph of the introduction.
>
> 3. We disagree that the experimental improvement of GLAM is limited, and point the reviewer to the induced sparsity shown in Table 2 (shown as % Edges Removed). This degree of sparsity has not yet been induced in these datasets while preserving the achieved SOTA classification performance. As such, we view these results as indicating, in addition to the efficacy of the GLAM layer, that there may be more redundant edges in these datasets than previously thought. Please see an updated Section 6.2, where we have included a summary of the above rebuttal to clarify our points.
>
> Thank you again for your careful review, and please let us know if you have any remaining questions.

---

### Decision · Program_Chairs · 2023-01-20

**Decision:**

Reject

**Justification For Why Not Higher Score:**

It is a unanimous reject by all reviewers, and the author responses did not add new experiments or address the concerns.

**Justification For Why Not Lower Score:**

N/A

**Metareview: Summary, Strengths And Weaknesses:**

This paper studies the graph structure learning problem where noisy edges are to be removed to faciliate node embedding learning. It proposes that structure learning and node embedding should be separated as they conflict with each other, which is generally appreciated by the reviewers and me as a good point. However, the current paper suffers from limited evaluation, insignificant empirical improvement, lack of deeper theoretical insights, etc. All reviewers suggest rejecting the paper.